https://doi.org/10.1038/s42003-022-03781-3　　OPEN
# Plant hairy roots for the production of extracellular vesicles with antitumor bioactivity

Eleonora Boccia[1,7], Mariaevelina Alfieri[1,2,7], Raffaella Belvedere[1], Valentina Santoro[1], Marianna Colella[1], Pasquale Del Gaudio[1], Maria Moros[3,4], Fabrizio Dal Piaz [5,6], Antonello Petrella [1], Antonietta Leone[1] & Alfredo Ambrosone [1✉]

Plant extracellular vesicles (EVs) concentrate and deliver different types of bioactive molecules in human cells and are excellent candidates for a next-generation drug delivery system. However, the lack of standard protocols for plant EV production and the natural variations of their biomolecular cargo pose serious limitation to their use as therapeutics. To overcome these issues, we set up a versatile and standardized procedure to purify plant EVs from hairy root (HR) cultures, a versatile biotechnological system, already successfully employed as source of bioactive molecules with pharmaceutical and nutraceutical relevance. Herewith, we report that HR of *Salvia dominica* represent an excellent platform for the production of plant EVs. In particular, EVs derived from *S. dominica* HRs are small round-shaped vesicles carrying typical EV-associated proteins such as cytoskeletal components, chaperon proteins and integral membrane proteins including the tetraspanin TET-7. Interestingly, the HR-derived EVs showed selective and strong pro-apoptotic activity in pancreatic and mammary cancer cells. These results reveal that plant hairy roots may be considered a new promising tool in plant biotechnology for the production of extracellular vesicles for human health.

[1] Department of Pharmacy, University of Salerno, 84084 Fisciano, Italy. [2] Clinical Pathology, Pausilipon Hospital, A.O.R.N Santobono-Pausilipon, 80123 Naples, Italy. [3] Instituto de Nanociencia y Materiales de Aragón (INMA), CSIC-Universidad de Zaragoza, Zaragoza, Spain. [4] Biomedical Research Networking Center in Bioengineering, Biomaterials and Nanomedicine (CIBER-BBN), Madrid, Spain. [5] Department of Medicine, Surgery and Dentistry "Scuola Medica Salernitana", University of Salerno, 84081 Baronissi, Italy. [6] Operative Unit of Clinical Pharmacology, University Hospital "San Giovanni di Dio e Ruggi d'Aragona", 84131 Salerno, Italy. [7] These authors contributed equally: Eleonora Boccia, Mariaevelina Alfieri. ✉email: aambrosone@unisa.it

Extracellular vesicles have been described in all three domains of the tree of life, suggesting that vesicular transport is a universal biological process through which cells exchange important biomolecules and genetic information, thus establishing an intercellular communication in the same or even between different organisms[1,2]. The International Society of Extracellular Vesicles (ISEV) consensus recommends using the term extracellular vesicles for particles, delimited by a lipid bilayer and which cannot replicate, naturally released from cells[3].

Plant-derived nano and microvesicles have being surprisingly almost neglected for a long time. In fact, nano-sized vesicles in plants have been observed in the '60s[4,5], earlier than animal exosomes[6,7], but their biological role remained largely unexplored. In the last decade, distinct classes of plant-derived nano and microvesicles (PDVs), including extracellular vesicles (EVs) have been obtained from tissues, organs, apoplastic fluids, and juices of several plant species.

Some of these studies have shown that PDVs are excellent nanocarriers able to transport bioactive molecules, including lipids, proteins, small-non coding RNAs, and other metabolites to target cells, driving cell-cell communication, even between different organisms and species. Proteomic studies have demonstrated that PDVs contain typical EV-associated proteins together with numerous proteins with antifungal, antimicrobial and hydrolytic activities, suggesting that EVs are important components of the innate immune system in plants and participate in cell wall organization[8–11].

Besides their physiological role in plants, it has been recently observed that PDVs may provide beneficial effects for human health[12,13]. For instance, PDVs derived from edible plants such as grapes, grapefruit and ginger have been demonstrated to contribute to the maintenance of the intestinal tissue homeostasis and repair mechanisms in animal models, favouring the process of renewal of enterocytes and alleviating inflammation[14–16]. Citrus-derived vesicles, especially those obtained from lemon and grapefruits, have shown antineoplastic effects by inhibiting cancer cells proliferation through the activation of the programmed cell death mechanisms both in vitro and in vivo[17,18]. Also, a few clinical studies have been started to assess the therapeutic role of the plant-derived nano and microvesicles in the clinical practice (ClinicalTrials.gov Identifier: NCT01668849; ClinicalTrials.gov Identifier: NCT01294072). More recently, the preliminary results of a pilot open label study demonstrated that EVs derived from *C. limon* juice exert positive effects reducing waist circumference and LDL cholesterol in healthy subjects[19].

Despite their promising potential, the safe and proper use of plant EVs for biomedical purposes is still limited due to the lack of standardization during the EV isolation and characterization. For instance, due to the natural diversity of plant-derived nano and microvesicles, their cargo and bioactivity may change substantially according to the sources (e.g., plant species, tissue and organs), as well as to the plant physio-pathological conditions. PDV availability and characteristics are also affected by the production seasons and accessibility of plant materials.

Moreover, the current PDV purification procedures are mostly based on destructive methods of tissues and organs which do not allow discriminating between true extracellular vesicles and other types of small vesicles released or artificially created after cell destruction[20]. This also contributes to the lack of a clear classification of plant-derived vesicles. Therefore, the development of a plant-based biotechnology system ensuring a standardised purification of true extracellular vesicles with a conserved biomolecular cargo and reproducible bioactivity is highly demanded.

As proof of concept, in this work we focused on setting up a reliable and reproducible procedure of purification and characterization of extracellular vesicles from the hairy root cultures of *Salvia dominica*, a medicinal plant known for the several biological activities of its extracts[21,22].

Hairy roots (HR) are neoplastic growth that originates from the wounding site of plants infected by *Agrobacterium rhizogenes*, a soil symbiotic bacterium currently taxonomically renamed *Rhizobium rhizogenes*. During the infection mechanism, specific bacterial DNA fragments (T-DNA) transfer from its root-inducing plasmid into the plant genome, stimulating the differentiation of hairy roots in many dicotyledonous plants[23].

Nowadays HR cultures have become major and versatile plant biotechnological tools attracting the interest of different industries in the pharmaceutical, nutraceutical and cosmetics fields[24]. Compared to other plant cell cultures, the hairy roots are stable under hormone-free culture conditions and offer additional advantages such as fast growth, low doubling time, ease of maintenance, absence of pathogen contaminants as well as the ability to synthesize a wide range of natural chemical compounds. Besides, strategies of metabolic engineering to control the biosynthetic pathways and to push the metabolic flux toward the desired compounds are viable routes to fully harness the potential applications of hairy root cultures[25,26]. Hairy roots obtained from many plant species have also been used to produce recombinant proteins[27–29] (e.g., human acetylcholinesterase, murine interleukin, human interferon alpha-2b, human erythropoietin) and specialized metabolites such as camptothecin, naphthoquinones, stilbenes, tanshinones, anthocyanins and many others[30–35].

Herein, we report the purification of round-shaped extracellular vesicles by differential ultracentrifugation from *S. dominica* HR-conditioned medium. They share biophysical characteristics and a pattern of associated proteins as previously described in animal and plant EVs, such as actin, chaperones and many integral components of plasma membranes. We also demonstrated that *S. dominica* HR-derived EVs have a selective and strong pro-apoptotic activity in pancreatic and breast cancer cells. The results of this work pave the way for the exploitation of hairy roots as novel biotechnological platforms for the standardised production of plant bioactive EVs to be used as pharmaceuticals and nutraceuticals.

## Results

**Hairy root induction and characterization**. Hairy roots (HRs) were obtained by infecting leaf explants excised from 20-days old *S. dominica* plantlets with *Agrobacterium rhizogenes* (Fig. 1a, d). Four week-old healthy and differentiated HRs were transferred into a sterile flask containing the liquid medium without phytohormones and grown in the dark under continuous agitation (Fig. 1e). The HRs were characterized for the stable genomic integration of the T-DNA region of the Root Inducing plasmid (pRI) of *A. rhizogenes* in HRs by *rolB* gene amplification using a polymerase chain reaction (PCR) (Fig. 1f). Elimination of *Agrobacterium* in the HR cultures after antibiotic treatment (Fig. S1) was verified by amplification of *virD2* virulence gene, carried by the T-helper plasmid of *A. rhizogenes*, which is not transferred into the plant genome. Only stable and *Agrobacterium*-free *S. dominica* HR cultures were then used to isolate EVs from their conditioned media, as schematised in Fig. 1g.

**Isolation and biophysical characterization of HR-derived EVs**. EVs were isolated by differential ultracentrifugation from the conditioned medium (~40 mL) collected from 10-day freshly inoculated HR cultures, which yielded an average of 2 grams (fresh weight) of HRs. Particle size distribution and morphology of HR-derived extracellular vesicles were characterized by different techniques. Dynamic light scattering (DLS) analyses revealed that EVs ranged in size prevalently from 140 to 210 nm (Fig. 2a) with the major peak for the particle hydrodynamic

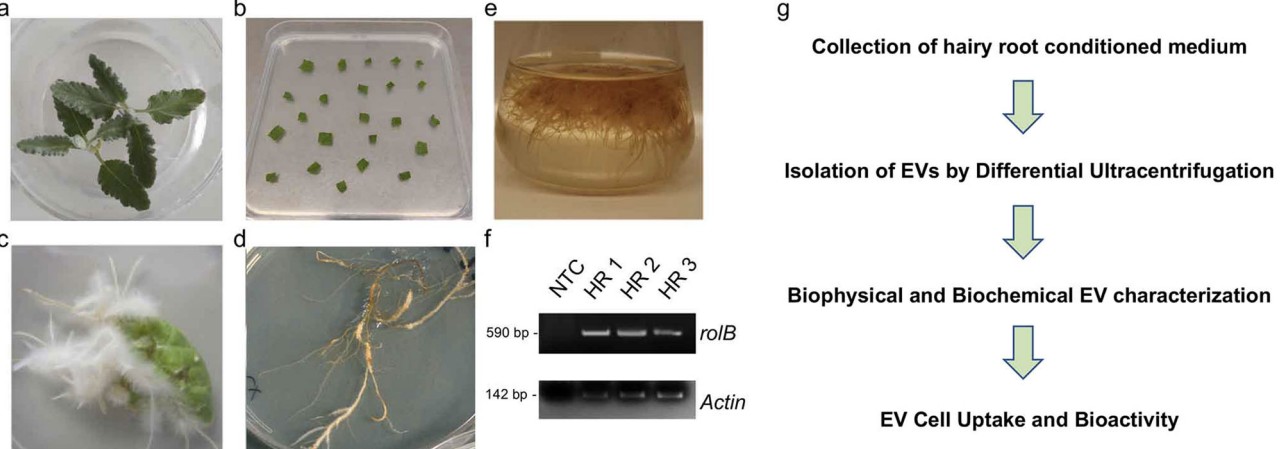

**Fig. 1 Experimental design for the production and characterization of *S. dominica* hairy root derived EVs. a** Plantlets of *Salvia dominica* grown in vitro. **b** Sterile leaf sections infected by *A. rhizogenes*. **c** Hairy roots (HRs) development from infected leaves. **d** Subcultures of excised hairy roots in solid hormone free medium. **e** HR in the hormone-free liquid MS medium. **f** Stable integration of the *rolB* gene in HRs assessed by polymerase chain reaction (PCR) in three independent transformation events. Amplicon length: 451 base pairs (bp) and 142 bp for *rolB* and *actin* genes, respectively. NTC, no template control. **g** Workflow planned for the purification and characterization steps of extracellular vesicles released by *S. dominica* HRs.

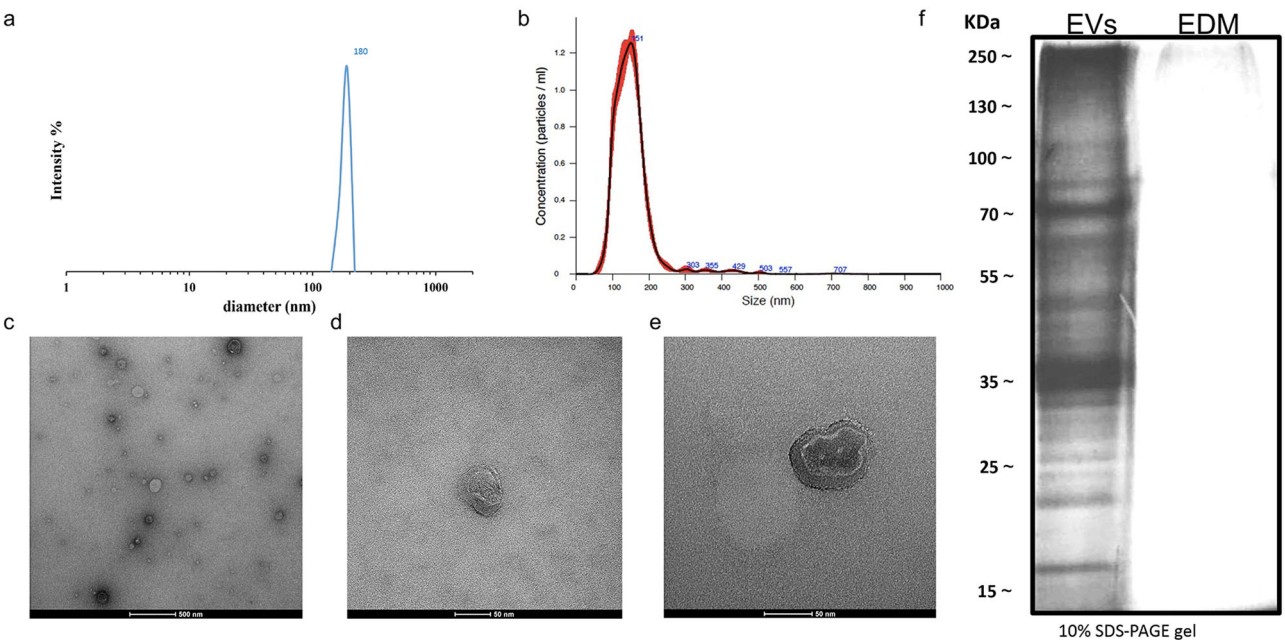

**Fig. 2 Biophysical characterization of HR-derived EVs. a** EVs size distribution curve obtained by dynamic light scattering (DLS). **b** Nanoparticle Tracking Analysis (NTA) measurements show the distribution of the EV size with the major peak at 151 nm. **c** TEM image of isolated vesicles exhibiting round-shaped morphology. **d, e** Close-up view images of individual HR-derived EVs. **f** Protein pattern of HR-derived EVs and EV depleted-medium (EDM) visualized by SDS-PAGE and silver staining. Silver staining of EDM did not show detectable proteins. Scale bars: 500 nm in **c**, 50 nm in d, 50 nm in **e**.

diameter at ~180 nm. As detergents are able to solubilize and destroy lipid bilayers of EVs, sensitivity to detergents has been used as a method to validate the vesicular nature of the isolated nanostructures[36,37]. DLS analysis of HR-derived EV preparations after 1 h incubation with 1% sodium dodecyl sulphate (SDS) showed the destruction of HR-derived EVs confirming that they are lipid membrane enclosed vesicles (Fig. S2).

Complementary approaches, such as Nanoparticle Tracking Analysis (NTA) and electron microscopy confirmed these measurements. NTA is a powerful technique that combines both laser light scattering and Brownian motion in order to characterize the hydrodynamic diameter and concentration of particles, including extracellular vesicles in liquid suspensions.

Also, NTA provides a better peak resolution than DLS offering an accurate size estimation of heterogeneous EV populations[38,39]. NTA revealed that 81.8% of detectable particles were within the size range of 100–200 nm with a mean diameter of around 153 nm+/− 1.8 nm. We noticed also the presence of nanovesicles (about 9% of the total particles analysed) with a diameter smaller than 100 nm (Fig. 2b and Fig. S3), not found in DLS measurements. Moreover, NTA detected the presence of particles with a hydrodynamic diameter higher that 250 nm (approx. 4%) probably generated by the formation of EV aggregates. According to NTA estimation, EV mean concentration was 1.89e + 09+/− 8.6e + 07 particle/mL. Transmission Electron microscopy (TEM) was used for wide-field and close-up imaging of EVs, respectively.

TEM images revealed the presence of intact round-shaped EVs, heterogeneous in size and morphology (Fig. 2c–e), mostly ranging between 70 nm and 160 nm, as expected.

The sodium dodecyl sulphate-polyacrylamide gel electrophoresis (SDS-PAGE) of *S. dominica* EVs showed a characteristic and reproducible protein pattern (Fig. 2f), indicating that the purification method does not affect the EV cargo stability and integrity.

Finally, any potential *A. rhizogenes* contamination in the EV preparations was excluded by PCR amplification of *rolB, rolC* and *virD2* genes as reported in Fig. S4.

**Proteome analysis of hairy root-derived extracellular vesicles**. EVs carry a distinct repertoire of luminal and membrane proteins implicated in their formation, release and cell uptake together with numerous molecular signal components able to control biological functions in the recipient cells. In order to provide a detailed protein characterization of HR-derived EVs and identify typical EV-associated biomarkers, we performed a classical gel-based proteomic approach in three distinct preparations of HR-derived EVs. To obtain a list of proteins enriched in the EV preparations, we also characterised the proteins present in the pellet fractions after low-speed centrifugation steps and subtracted them to the EV protein dataset. As the information available in the database of *S. dominica* proteome is rather limited, proteins were identified against the Green Plant (Viridiplantae) protein database.

The proteins commonly found in the EV preparations and in the non-vesicular fractions, reported as supplementary data 1, were not considered further in this work. Conversely, in two replicate analyses performed on two independent experiments, we found 143 proteins exclusively associated to *S. dominica* HR-derived EVs (table S1). The proteins with the highest Mascot score among these are listed in Table 1.

Gene Ontology term enrichment of cellular components showed the presence of numerous proteins in the categories "membrane", "plant -type cell wall organization" and "extracellular components" (Fig. S5a) confirming again the vesicular nature of EVs released from HRs.

Moreover, a wide range of molecular functions can be covered by these proteins (Fig. S5b). Remarkably, we found the presence of a large number of ribonucleotide binding proteins (Fig. S5c) that may be involved in the nucleic acid packaging and extracellular transport, relevant for the intercellular and inter-kingdom signalling role of EVs[2,40–42].

Interestingly, proteomic data reported in Table 1, table S1, and in Supplementary data 1 evidenced also that *S. dominica* EV carry many proteins frequently found in mammalian extracellular vesicles according to Vesiclepedia database[43], such as cytoskeletal components (actin, tubulin, kinesin proteins), chaperon proteins (HSP70s, HSP 80 and HSP90), glycolytic enzymes (enolases and glyceraldehyde-3-phosphate dehydrogenases), annexin-like protein and elongation factors. It is worthy to evidence that HR-derived EVs carry tetraspanin-7 (TET7), a protein with high homology to *A. thaliana* TET8, which is considered a specific plant exosomal marker.

Finally, numerous integral components of membranes, commonly described in the vesicles isolated in animals and plant species were also found in HR-derived EV protein dataset. Among them, we identified an ATP synthase subunits, a LRR receptor-like serine/threonine-protein kinase HSL2, the Aquaporin TIP3-1, the probable phospholipid-transporting ATPase 8, the Ras-related proteins RABA4a and Rab11A.

***Salvia dominica* HR-derived Extracellular Vesicles have anti-tumor activities**. Antibacterial, antioxidant, antidiabetic, cardiovascular and antitumor activities of natural products obtained from *Salvia* species have been largely documented[44]. In particular, *S. dominica* extracts exhibited antiproliferative activity against different cell tumor lines[21,22]. Hence, we evaluated whether EVs released from HR cultures of this *Salvia* species might also have anticancer activity.

Firstly, we examined the EV cell uptake in the HaCaT cells (human keratinocytes), already employed by our group as non-malignant cells to test anticancer activity of several plant-derived

**Table 1 List of the 20 vesicle proteins showing the highest score in the Mascot-based bioinformatics analysis carried out on proteomic data.**

| Accession | Score | Mass | Num. of significant matches | Num. of significant sequences | Description |
|---|---|---|---|---|---|
| G3PC_DIACA | 475 | 37105 | 28 | 11 | Glyceraldehyde-3-phosphate dehydrogenase |
| G3PC2_ORYSJ | 389 | 36921 | 25 | 11 | Glyceraldehyde-3-phosphate dehydrogenase 2 |
| EF1A1_DAUCA | 366 | 49613 | 17 | 8 | Elongation factor 1-alpha |
| HSP70_SOYBN | 309 | 71291 | 17 | 9 | Heat shock 70 kDa protein |
| METE1_ORYSJ | 307 | 84874 | 12 | 5 | 5-methyltetrahydropteroyltriglutamate--homocysteine methyltransferase 1 |
| BIP1_ARATH | 292 | 73869 | 15 | 8 | Heat shock 70 kDa protein BIP1 |
| ALF2_PEA | 280 | 38638 | 10 | 6 | Fructose-bisphosphate aldolase |
| METE3_ARATH | 249 | 90993 | 10 | 4 | 5-methyltetrahydropteroyltriglutamate--homocysteine methyltransferase 3 |
| API3_SOLTU | 226 | 18909 | 12 | 6 | Aspartic protease inhibitor 3 |
| PGKH2_ARATH | 208 | 50022 | 6 | 4 | Phosphoglycerate kinase 2 |
| 1433B_TOBAC | 207 | 29037 | 16 | 6 | 14-3-3-like protein B |
| 14337_ARATH | 201 | 29920 | 16 | 6 | 14-3-3-like protein GF14 nu |
| CDC48_SOYBN | 190 | 90512 | 10 | 6 | Cell division cycle protein 48 homolog |
| ATPAM_SOYBN | 185 | 55581 | 7 | 5 | ATP synthase subunit alpha |
| LOX1_LENCU | 177 | 96807 | 7 | 2 | Linoleate 9S-lipoxygenase |
| 14331_SOLTU | 174 | 29491 | 16 | 6 | 14-3-3-like protein |
| MDHM_CITLA | 164 | 36406 | 6 | 2 | Malate dehydrogenase, mitochondrial |
| ENPL_CATRO | 154 | 93605 | 5 | 3 | Endoplasmin homolog |
| VCL1_PEA | 152 | 14031 | 9 | 4 | Vicilin, 14 kDa component |
| ENO2_ARATH | 152 | 47974 | 7 | 3 | Bifunctional enolase 2/transcriptional activator |

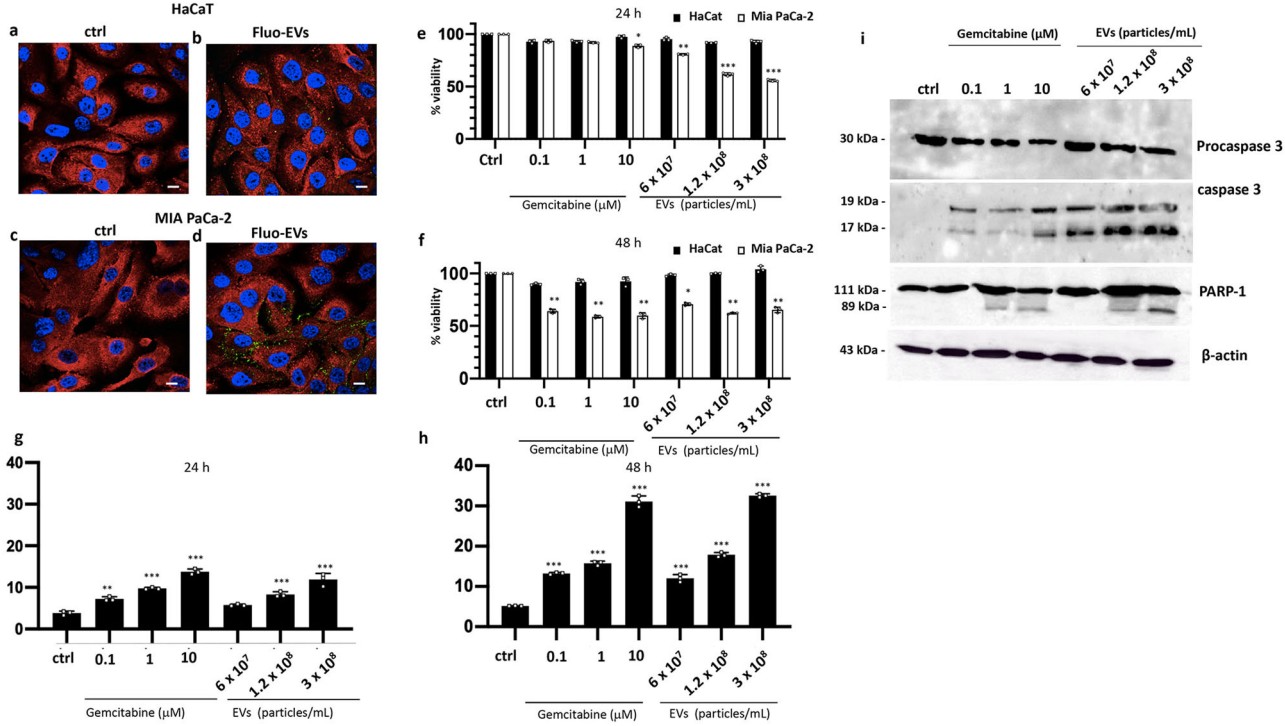

**Fig. 3 Uptake and biological effects of HR-derived EVs in human control and cancer cell lines. a–d** Confocal analysis of HaCaT and MIA PaCa-2 cells treated with Bodipy-stained EVs (Fluo-EVs) (1.2 × 10⁸ particles/mL) for 24 h. Cells have been stained for annexin A1 protein in red. Nuclei were stained with DAPI. Magnification 63×/1.4 numerical aperture. Scale bar = 100 μm. **e, f** MTT colorimetric assay on HaCaT and MIA PaCa-2 cells after 24 h and 48 h of EV treatments, respectively. Absorbance relative to controls was used to determine the percentage of cells treated with varying concentrations of gemcitabine and EVs. **g, h** Analyses of apoptotic cells by cytofluorimetric assay in gemcitabine- and EVs-treated MIA PaCa-2 cells upon 24 h and 48 h exposure. The values reported in the graphs are the mean ± SD from at least 3 independent experiments performed in technical triplicates. The asterisks denote significant differences between treatments and untreated controls (*$p < 0.05$; **$p < 0.01$; ***$p < 0.001$) according to Student's $t$-test. **i** Western blot analyses of protein extracts from cells treated for 24 h with different concentrations of gemcitabine and EVs. Levels of cleaved proteins involved in the apoptosis (Procaspase 3 and PARP-1) were evaluated. β—actin was used to check equal loading of protein extracts.

vesicles[18], and MIA PaCa-2 pancreatic carcinoma cells, characterised by high invasiveness and drug-resistance[45]. To this aim, the cell lines were incubated for 24 h with HR-derived EVs (1.2 × 10⁸ particles/mL) labelled with the lipophilic fluorescence dye Bodipy FL (Fluo-EVs). A confocal microscopy analysis showed that Fluo-EVs mostly enter and accumulate into the cytoplasm, preferentially in cancer cells (Fig. 3a–d). Longer incubation times up to 48 h did not result in an increased accumulation of Fluo-EVs in cancer cells (Fig. S6). To check the stability of EVs in different culture conditions, we evaluated the uptake levels in cells cultured in fetal bovine serum (FBS)-free media. Confocal images confirmed that the Fluo-EV internalization levels were comparable to those observed in cells grown in complete culture medium (Fig. S7), indicating that HR-derived EVs stability and uptake are not affected by the presence of exogenous FBS proteins that could cause extensive EV aggregation during the incubation.

Subsequently, we evaluated the impact of HR-derived EVs on the growth of HaCaT and MIA PaCa-2 cell lines. In particular, cells were incubated for 24 h and 48 h with increasing concentrations of EVs (from 6 × 10⁷ to 3 × 10⁸ particle/mL). EV effects were compared to those of gemcitabine, a chemotherapeutic drug widely used for pancreatic cancer treatment known to inhibit DNA synthesis and promote cell apoptosis[46,47]. Additionally, both cell lines were incubated with heat-denatured EV preparations, which have been previously used as negative controls[48–50]. Cell viability was determined by the tetrazolium-based cytotoxicity assay (MTT). As shown in Fig. 3e–f, *S. dominica* EVs did not alter HaCaT viability after 24 h and 48 h treatment, indicating a safe profile for non-cancer cells. Conversely, HR-derived EVs severely affected the

cell viability of MIA PaCa-2 cell line in a dose-dependent manner with the highest cell viability reduction (>40%) when used at a concentration of 3 × 10⁸ EVs/mL for 24 h (Fig. 3e). Under these conditions the cytotoxic effects of EVs in cancer cells were significantly higher than those in cells treated with gemcitabine. Longer incubation with EVs did not reduce further the cell viability of MIA PaCa-2 cells (Fig. 3f). Heat-denatured EVs induced only non-significant slight fluctuations of cancer cell viability (Fig. S8). Taken together, these data prove that intact HR-derived EVs exert selective and early cytotoxic effects in cancer cells.

To investigate deeper the bioactivity of HR-derived EVs on tumor cells, MIA PaCa-2 cells exposed to HR-derived EVs for 24 h and 48 h were analysed by fluorescence-activated cell sorting (FACS) as reported in Fig. 3g–h, respectively. FACS analyses showed that the treatment with HR-derived EVs induces a dose-dependent increase of cell apoptosis compared to the control cell in tumor cells, but not in HaCat cells (Fig. S9). Remarkably, we found that pro-apoptotic activity of EV treatments was comparable to that observed in cancer cells treated with different concentrations of gemcitabine. Western blot analyses (Fig. 3i and Fig. S12) were carried out to further confirm at molecular level the programmed cell death execution through the activation of procaspase-3 and subsequent cleavage of poly (ADP-ribose) polymerase-1 (PARP-1), which is one of the several known cellular substrates of caspases. In particular we found that the 32 KDa band of pro-caspase 3 was cleaved in EV-treated MIA PaCa-2 cells in a dose dependent manner. As expected, partial degradation of PARP1 was detected after 24 h EV treatment. To confirm the antitumor activity of *S. dominica* HR EVs, we evaluated their effects also in MCF-7 breast

cancer cell line. As reported in Figs. S10a and S10b, 24 h and 48 h EV treatments reduced up to 60% the MCF-viability and promoted programmed cell death in a dose-dependent manner, respectively. Selective procaspase-3 and PARP-1 cleavage in MCF7 cells confirmed the activation of the apoptotic molecular cascade upon 24 h EV treatment (figs. S10c and S13). As expected, heat-denatured EV did not affect MCF-7 viability (Fig. S11).

Altogether, these data clearly indicate that *S. dominica* HR-derived EVs play a positive role in restoring programmed death in cancer cells, likely triggered through the localised release of anticancer molecules carried by EVs.

## Discussion

Most of the plant-derived nano and microvesicles so far isolated have been obtained through destructive methods that lead to the purification of heterogeneous vesicle populations. In these preparations, EVs and intracellular nano or microvesicles derived from the rupture of plant tissues and organs are mixed together. Not least, artificial vesicles unintentionally produced during the invasive extraction procedures may contaminate the EV samples[20]. Actually, only a handful of works have described true EVs collected from extracellular apoplastic fluids of plants[8,10,51–54]. More recently, we reported that tomato plants, grown in hydroponic culture, release EVs with antifungal properties in the surrounding environment through the root system[11]. These previous findings inspired the present study aimed at setting up a reliable biotechnological platform for the production and purification of biologically active plant EVs based on hairy root (HR) technology.

In particular, to the best of our knowledge here we demonstrated for the first time that HRs obtained from *Salvia dominica* secrete round-shaped extracellular vesicles ranging in size prevalently between 100–200 nm. The lipidic membranous nature of HR-derived vesicles was confirmed through the staining with the specific hydrophobic dyes BODIPY FL able to bind lipid membranes and through the detergent SDS sensitivity assay. In addition, proteomic analysis of the HR-derived EVs revealed the presence of a consistent number of membrane and extracellular proteins and other reported EV-associated proteins. We also noticed in the HR-released EV proteome the absence of the category "endoplasmic reticulum" (Fig. S5A). According to a recent comparative proteomic study of plant-derived nanovesicles and small EVs in *A. thaliana*, this could be a distinctive feature of the true EV proteome[55].

*S. dominica* HR-released EVs cargo contains proteins such as elongation factors, heat shock proteins, glyceraldehyde 3 phosphate dehydrogenases and integral membrane proteins, also reported in the proteome of EVs isolated from important plant species[11,53,54]. Interestingly, HR-derived EVs also carry TET-7, a tetraspanin with high homology to the plant exosomal marker TET-8. Finally, the protein cargo of EVs released by *S. dominica* HRs is comparable also to those reported in a recent survey on the protein families most frequently found in plant EV proteomes[20]. This indicates that this set of proteins might be considered as reliable markers of plant EVs, although it is necessary to extend the analysis to EVs purified from a larger number of plant species, to establish a general consensus of the scientific community on bona fide protein markers of plant EVs.

Taken together these data indicate that HR-released vesicles are true plant EV, however the presence of a small number of vesicles generated by accidental rupture of soft structures (e.g. root hairs) in the growth phase cannot be completely excluded.

Remarkably, the size, the morphology and the protein cargo of HR-derived EVs significantly diverge from EVs that we have previously isolated from root exudates of hydroponically grown tomato plants, suggesting that plant hairy roots may release in the environment EVs with peculiar biophysical and biochemical characteristics compared to in vivo roots.

An overwhelming number of recent data has largely proved that the biological activity of plant EVs is strictly associated to their biomolecular cargo which contains high concentrations of bioactive molecules (e.g., proteins, specialised metabolites and small-non coding RNAs) packed in nano and microstructures and is due to their innate delivery function. For these peculiar features, plant EVs have attracted great attention as new candidates for the development of innovative therapeutics. Their potential biomedical applications have been supported in the last few years by a growing body of evidence that has shown how PDVs play an important role in the regulation of gut microbiome and exert important antioxidant, anti-inflammatory and anticancer properties[12,56,57]. For instance, it has been proved that lemon-derived vesicles were able to inhibit cell proliferation of tumor cell lines[17] in part by inhibiting the lipid metabolism of cancer cells[58]. Recently we have demonstrated that grapefruit-derived nano and microvesicles carry antioxidant enzymes and anticancer compounds conferring antiproliferative and pro-apoptotic activities in a melanoma cell line[18,59]. Moreover, PDVs have shown a low immunogenicity and can be virtually obtained at low cost from a considerable number of plant sources[15,60]. Despite their great promise, the standardisation of purification processes and biomedical applications of PDVs struggle to keep up. Data presented herewith have demonstrated that EVs are released from HRs of *S. dominica* and can be easily purified from the growth medium. This constitutes an important biotechnological advance for ensuring an easy, cost-effective, scalable and rapid purification of bioactive plant EVs. A HR-based production of plant EVs avoids invasive extraction procedures from plant tissue and organs and contributes also to achieve a standardised EV quality, abolishing the effects on PDVs variation of multiple factors, such as the developmental stages, the environmental conditions and genetic diversity of plant sources. At the same time it ensures the sterility of EV preparations required for biomedical applications.

In this work, we have used HRs of *Salvia dominica* as EV source. The genus *Salvia* is characterised by high biodiversity including over 900 species in the Lamiaceae family[61]. Besides their use in food consumption, many of these species are routinely used in traditional medicine to treat cardiovascular diseases and infections and are included in the official pharmacopeia lists of different countries. Most of the therapeutic effects of *Salvia* species are probably determined by the elevated content of poliphenols, flavonoids and terpenoids. Cell and organ cultures of several *Salvia* species have been set up to produce interesting compounds such as rosmarinic acid, aethiopinone and tanshinone[26,62–64].

Our data have shown that that HR-derived EVs of *S. dominica* possess a strong and selective pro-apoptotic activity in pancreatic and breast cancer cells with a safe profile for non-cancer cells. It is worthy to note that the antiproliferative and pro-apoptotic properties of HR-derived EVs in Mia PaCa-2 cells were comparable or even higher than those of gemcitabine, a chemotherapy drug used to treat several solid tumors. Further studies are necessary to establish which components of the EVs released from *S. dominica* HRs are responsible of this clear and strong antitumor activity we have observed in pancreatic and breast tumor cells. Although the therapeutic properties of *S. dominica* have not yet been fully elucidated, previous works have reported that the extract of this species may exert anticancer activity[21,22], in part mediated by sesterterpene lactones which interact with and inhibit the tubulin−tyrosine ligase (TTL), an enzyme involved in the tyrosination cycle of the C-terminal of tubulin[65].

Furthermore, we proved that fluorescently-labelled EVs are internalized and accumulate in the cytoplasm of keratinocytes and cancer cells within 24 h. This timing of internalization is similar to that observed in mammalian exosomes and other plant EVs, which is normally completed in a few hours. The mechanisms of plant EVs uptake, specific surface interactions and molecular players involved in mammalian cell/plant EV recognition are only marginally known. For instance, surface protein interactions between a mannose-specific binding protein II lectin and CD98 may play an important role as reported for the internalization of garlic-derived vesicles in human liver cancer cells[66]. Further studies are needed to determine whether and how specific HR-derived EV proteins or lipids drive the internalization processes in human cells.

Interestingly, the ability of HR-derived EVs to be taken up by human cells is a tremendous technological tool to concentrate and locally deliver one or more co-acting biomolecules, which ultimately may result in an expected high therapeutic efficacy. This advantage in using EVs for the delivery of therapeutics was clearly indicated by our finding that EVs disrupted by heat denaturation lose their bioactivity against cancer cells. A deep characterization of the biomolecular cargo of HR-derived EVs by metabolomic, lipidomic and transcriptomic analyses will be carried out for a more comprehensive understanding of EV effects in cancer cells or other recipient cells.

In conclusion, to the best of our knowledge, this study is the first report on the EV purification from a hairy root biotechnological platform. In particular, we demonstrated that hairy roots might be used as biofactories of extracellular vesicles with interesting biological properties to be explored in more depth in the nutraceutical and pharmaceutical sectors. It appears appealing the possibility to scale up the EV production starting from controlled biomass derived from plant cell and organ cultures, which respond better to the good manufacturing procedures and standardised quality requested by the pharmaceutical industry. Other biotechnological approaches, might be envisaged to optimize further the production of smart plant EVs from cell and tissue cultures, for example by elicitation or genetic engineering, aimed at increasing the final yield and/or customizing the EV cargo with specific bioactive natural molecules, which however require a deeper comprehension of plant EVs biogenesis.

## Methods

**Plant material and growth conditions.** *Salvia dominica* seeds were kindly provided by Ammar Bader (Department of Pharmacognosy, Umm Al-Qura University, Saudi Arabia). The seeds were surface sterilized with 70% (v/v) ethanol for 1 min and then with 2% (v/v) sodium hypochlorite solution for 10 min, rinsed five times with sterile distilled water for 1 min each and then sown under aseptic conditions in autoclaved Murashige & Skoog (MS) medium (pH 5.8) containing 30 g/L sucrose and 9 g/L agar. The plants were grown in vitro at 23 °C under a photoperiod of 8 hour dark and 16 hour light in a controlled growth chamber with a photosynthetically active radiation of 110 μmol m$^{-2}$ s$^{-1}$ at the leaf level.

**Hairy root cultures production and characterization.** The protocol to obtain hairy roots of *Salvia dominica* was adapted from our previous work[26]. Briefly, the *Agrobacterium rhizogenes* virulent strain ATCC 15834 obtained from a single colony was grown (OD$_{600}$ = 0.3–0.5) in Yeast Extract Broth (YEB) liquid medium (0.1% Yeast extract, 0.5% Beef extract, 0.5% Peptone, 0.5% Sucrose, 0.04% MgSO$_4$) at 26 °C. Before transformation, bacteria were collected from liquid cultures at 5000 rpm for 5 min and resuspended at a cell density OD$_{600}$ = 0.5 in YEB and were shaken for 1 h in the rotary shaker before inoculation. *S. dominica* sterile leaf explants from 20 day-old plants aseptically grown in vitro were submerged in the bacterial suspension for 30 min. Then, leaf explants were taken and dried under laminar flow hood for 15 minutes and co-cultivated at 26 °C for 3 days in MS30 solid medium and then transferred to fresh MS medium containing cefotaxime (CX, 100 mg/L) and kept in petri dish in the dark at 23 °C. After three weeks of several subcultures of infected leaves in media containing decreasing concentrations of CX (from 100 mg/L to 50 mg/L), hairy roots developing from the infected sites were individually excised, sub-cultured weekly in solid MS medium and kept for additional two weeks in the dark at 23 °C. Subsequently hairy roots

were transferred in hormone free liquid MS medium and maintained in aseptic condition (MS medium, CX 50 mg/L) at 26 °C in the dark in a shaking incubator (140 rpm). To confirm that the differentiated roots were HRs (and not adventitious roots) and that *A. rhizogenes* was efficiently eradicated, PCR analysis was carried out in order to check the presence of *rolB* and the absence of *virD2* genes, respectively. DNA purification of three independent hairy root lines and Polymerase Chain Reaction (PCR) were carried out by using Thermo Scientific *Phire* Plant Direct *PCR Kit* and the gene-specific primers: *Actin* (HM231319, forward primer: 5′-GGTGCCCTGAGGTCCTGTT-3′; reverse primer: 5′-GAGCCAC-CACTGAGGACAAT-3′; amplicon length: 142 bp); *rolB* (GenBank:X03433.1; forward primer: 5′- GATTCAACCATATCGGAGCG -3′; reverse primer: 5′-TGAGCATGTGTGCTGTTTTTGG -3′; amplicon length: 590 bp*); rolC* (MT514512.1; forward primer: 5′-CGACCTGTGTTCTCTTTTTCAAGC-3′; primer reverse: 5′-GCACTCGCCATGCCTCACCAACTCACC-3′; amplicon length: 514 bp); *VirD2* (*forward primer: 5′*-ATGCCCGATCGAGCTCAAGT; *reverse primer: 5′*-CCTGACCCAAACATCTCGGCT-3′; amplicon length: 317 bp).

PCR was performed using the following conditions: initial denaturation at 98 °C for 5′, followed by 40 cycles of denaturation at 98 °C for 15", annealing at 55 °C for 15" and extension at 72 °C for 45", and final extension at 72 °C for 5′. Then, PCR amplified products were analyzed on a 2% agarose gel.

**Isolation of extracellular vesicles.** Extracellular vesicles were isolated from the conditioned culture medium of *S. dominica* hairy roots (approximately 40 mL of medium for 2 grams of hairy roots) using differential centrifugation method. Overall, the medium was collected after 10 days of subculturing and used fresh for EV isolation. Alternatively, the conditioned media were stored in 50 mL tubes at –80 °C up to one month.

Briefly, the medium was initially centrifuged at 300 × g for 10 minutes and then at 2000 × g for 20 minutes to remove living or dead cells and cell debris. Afterwards, the supernatants were filtered using 0.45 μm pore size membrane filters (Merck Millipore) to remove contaminating apoptotic bodies, very small cell debris and organelles.

Additional low velocity centrifugation was performed using 50 mL Beckman polypropylene tubes with the Beckman Type 55.2 Ti fixed angle rotor (k-factor: 64) at 15,000 × g for 20 minutes at room temperature. Clarified conditioned medium was then centrifuged to pellet extracellular vesicles using thick wall polycarbonate tubes (Beckman Coulter, No. 355631) in a Beckman Coulter Optima™ L-90K ultracentrifuge at 100,000 × g $_{avg}$ at 4 °C for 3 hours with a Type 55.2 Ti rotor. The pellet was solubilised in 30 μl of MilliQ water and transferred in 1.5 mL tubes. After the isolation, EV were used fresh for subsequent analyses or stored at –80 °C avoiding freeze-thaw cycles.

**EV protein profiling by SDS-PAGE.** The quality of the vesicle samples and the reproducibility of EV isolation were controlled using sodium dodecyl sulphate polyacrylamide gel electrophoresis (SDS-PAGE). First, ten microliters of each samples (corresponding approximatively to 1.9e + 07 EVs) were electrophoretically separated under reducing conditions on polyacrylamide gel and then silver-stained as described by Shevchenko et al.[67] with small modifications. Silver staining has been conducted in all the EV preparations.

**Dynamic Light Scattering (DLS).** Size distribution of extracellular nanovesicles was measured by dynamic light scattering (DLS) using a Zetasizer Ver. 7.01, Malvern Instrument (Malvern, UK). Ten microliters of EV preparation were dispersed in deionized MilliQ water and the intensity of the scattered light was measured with a detector at 90° angle at room temperature. To prove the EV sensitivity to detergents, HR-derived EVs were incubated with 1% SDS and kept at room temperature for 1 h. Then the samples were analysed by DLS as above mentioned. Measurements were carried out in three different EV preparations.

**Nanoparticle Tracking Analysis (NTA).** The size distribution and concentration of HR-derived EVs were analysed by NTA using a NanoSight model NS300 equipped with a Blue405laser and a sCMOS camera (Malvern Instruments, Malvern, UK). The samples (15 μl) were diluted in 0.5 ml PBS and mixed well, and the diluted samples were then injected into the laser chamber. The following settings were used for data acquisition: camera level 9, acquisition time 30 s, and detection threshold 2. For each measurement, three videos were captured under the following conditions: cell temperature: 24 °C; Syringe speed: 50 μl/s. After capture, the videos have been analysed by the NanoSight Software NTA 3.4 with a detection threshold of 2. Three independent experiments were carried out.

**Transmission Electron Microscopy (TEM).** Before using a grid of carbon film supported on 300 mesh copper grid, a glow discharge was performed on it (30 s 15 mA). The samples were fixed using glutaraldehyde 2.5% in PBS and deposited on a parafilm. The grid was placed on the drop for 5 min, washed with distilled water for 1 min, and placed for 1 min in a drop of 3 microliters of 2% staining agent (Methylamine Vanadate, NanoVan) and diluted in H$_2$O before washing it with water for 1 min. TEM images were collected using a FEI Tecnai T20 (FEI Europe, Eindhoven, Netherlands) in the Laboratorio de Microscopias Avanzadas at

Universidad de Zaragoza (Spain). TEM observations have been carried out in three independent EV preparations.

**Proteomic and Bioinformatic analyses**. Protein extracts from purified HR-derived EVs and from pellet fractions collected at low-speed centrifugation were separated by 12% SDS-PAGE. Resulting gels were divided into 10 pieces, and each underwent a trypsin in gel digestion procedure. The obtained peptide mixtures were analysed on a Orbitrap Q-Exactive Classic Mass Spectrometer (Thermo Fisher Scientific) coupled with a nanoUltimate300 UHPLC system (Thermo Fisher Scientific). Peptide separation was performed on a capillary EASY-Spray PepMap column (0.075 mm × 50 mm, 2 µm, Thermo Fisher Scientific) using aqueous 0.1% formic acid (A) and $CH_3CN$ containing 0.1% formic acid (B) as mobile phases and a linear gradient from 3 to 40% of B in 45 minutes and a 300 nL·min−1 flow rate. Mass spectra were acquired over an m/z range from 375 to 1500. To achieve sequence confirmation, MS and MS/MS data underwent Mascot software (v2.5, Matrix Science) analysis using the non- redundant Data Bank UniprotKB/Swiss-Prot (Release 2021_03), limited to green plants proteins (Viridiplantae). Parameter sets were: trypsin cleavage; carbamidomethylation of cysteine as a fixed modification and methionine oxidation as a variable modification; a maximum of two missed cleavages; false discovery rate (FDR), calculated by searching the decoy database, 0.05. Proteomic analyses were performed in duplicate on three different vesicle preparations and only the proteins commonly identified in all the experiments were considered. Gene ontology analysis of identified proteins was carried out using the specific tool in UniProt Knowledgebase (UniProtKB; http://www.uniprot.org).

**Human cell culture**. MIA PaCa-2 (CRL-1420™) are immortalized epithelial cells of human pancreatic carcinoma. They were purchased from ATCC (Manassas, VA, USA) and cultured as reported[68]. HaCaT cells are human immortalized keratinocytes. They were purchased from CLS Cell Lines Service GmbH (Germany) and maintained in Dulbecco's modified Eagle's medium (DMEM) with 10% fetal bovine serum (FBS). MCF-7 (HTB-22™) are human breast cancer cells and were purchased from ATCC (Manassas, VA, USA). Both cell lines were maintained in Dulbecco's modified Eagle's medium (DMEM) with 10% fetal bovine serum (FBS).

All the media were supplemented with antibiotics (10,000 U/ml penicillin and 10 mg/ml streptomycin), cells were stained at 37 °C in 5% $CO_2$ −95% air humidified atmosphere and were serially passed at 70–80% confluence.

**MTT assay**. HaCaT, MIA PaCa-2 and MCF-7 cells were seeded in $15 \times 10^3$ cells/well in triplicate in 96 well-plates and incubated for 24 and 48 h with freshly prepared *S. dominica* EVs at a three different concentrations, namely $6 \times 10^7$, $1.2 \times 10^8$ and $3 \times 10^8$ particles/mL. Cells treated with heat-inactivated EVs were used as negative controls. To this aim, EV preparations were kept at 95 °C for 1 h, cooled down at room temperature for 2 h and then administered to the cell cultures.

As reported in Bizzarro et al.[69], at the ends of the selected experimental times, 3-(4,5-dimethylthiazol-2-yl)-2,5-diphenyltetrazolium bromide (MTT) stock solution (5 mg/ml) was added to each well (25 µl per 100 µl medium), and plates were incubated at 37 °C for 3 h. Cells were then lysed and the dark blue formazan crystals dissolved with 100 µl of a solution containing 50% (v/v) N, N-dimethylformamide, 20% (w/v) SDS with an adjusted pH of 4.5. The optical density (OD) of each well was measured in a microplate spectrophotometer (Titertek Multiskan MCC/340) equipped with a 620 nm filter. The viability of cells in response to the treatments with gemcitabine, HR-derived EVs and heat-denatured EVs was calculated as: % viable cells = [OD (550 nm–690 nm) treatments/OD (550 nm–690 nm) negative control] × 100. Values were presented as means ± SD of three independent experiments carried out in triplicate.

**Cell uptake and confocal microscopy**. EVs were labelled with BODIPY® FL N-(2-aminoethyl)maleimide. Briefly, 10 µL of Bodipy dye was added to 13.5 mL of *S. dominica* HR conditioned medium and mixed for 30 sec. After 5 min incubation at room temperature, EVs were isolated by ultracentrifugation as described before. After 24 and 48 h EV treatment, MIA PaCa-2 cells were harvested and used for confocal analysis performed as reported in a previous work[70]. Briefly, cells were fixed in p-formaldehyde (4% v/v in PBS; Lonza; Basel, Switzerland), were permeabilized with Triton X-100 (0.4% v/v in PBS; Lonza; Basel, Switzerland), blocked with goat serum (20% v/v in PBS; Lonza; Basel, Switzerland) and then incubated with anti-annexin A1 antibody (rabbit polyclonal; 1:100; Thermo Fisher Scientific; Waltham, MA, USA), overnight at 4 °C. After two washing steps, cells were incubated with anti-rabbit AlexaFluor 555 (1:500; Thermo Fisher Scientific; Waltham, MA, USA) for 2 h at RT in the dark. To detect the nuclei, 4′,6-diamidino-2-phenylindole (DAPI, 1:1000) was used. Samples were vertically scanned from the bottom of the coverslip with a total depth of 5 µm and a 63X (1.40 NA) Plan-Apochromat oil-immersion objective. Images and scale bars were generated with Zeiss ZEN Confocal Software (Carl Zeiss MicroImaging GmbH) and presented as single stack. Confocal microscopy analyses were carried out in three independent experiments.

**Apoptosis analysis by Flow Cytometry**. The percentage of hypodiploid nuclei was estimated by propidium iodide (PI) incorporation into permeabilized cells, as described by Belvedere et al[68]. Briefly, cancer cells were harvested after treatment with *S. dominica* EVs and incubated with a PI solution (0.1% sodium citrate, 0.1% Triton X-100) and 50 g/ mL of PI (Sigma-Aldrich, St. Louis, MO, USA), for 30 min at room temperature. Data from 10,000 events for each sample were collected by a FACS Calibur flow cytometer (Becton Dickinson, Franklin Lakes, NJ, USA) using Cell Quest evaluation program. FACS analyses were conducted in technical triplicate at least in three independent biological experiments.

**Western blot analysis**. Cells were lysed in a RIPA buffer containing 10 mM Tris HCl, pH 7.5, NaCl 150 mM, and Triton 1% supplemented with Protease Inhibitor Cocktails (Sigma Aldrich) and the protein content was measured by a colorimetric assay (BIO-RAD) as described in[71]. Protein cell lysates (20 µg) were separated by SDS-PAGE under denaturing conditions, transferred onto a nitrocellulose filter (BIO-RAD), blocked with 5% milk and probed with primary antibodies: mouse anti-PARP1 (1:500), rabbit anti-procaspase-3 (1:500) and mouse anti-βactin (1:1000) (Santa Cruz Biotechnology). After primary antibody incubation, the blots were washed and incubated with horseradish peroxidase-conjugated anti-mouse and anti-rabbit secondary antibodies (Thermo Fisher Scientific). The proteins recognized by Abs were detected by ECL (Amersham biosciences) and exposed to Las4000 (GE Healthcare Life Sciences) and the relative band intensities were determined using ImageQuant software (GE Healthcare Life Sciences). Pre-stained protein markers (Thermo Fisher Scientific) were used as molecular size standards. All experiments were performed in triplicate.

**Statistics and reproducibility**. Sample size and number of replicates for each experiment were described in figure legends, where possible. Otherwise these data were properly reported throughout the materials and methods section. Error bars in the graphical data represent standard deviations. A Student's t-test was used for statistical analysis, and P-values ≤0.05 (*), ≤0.01 (**) and ≤0.001 (***) were considered to be statistically significant.

**Reporting summary**. Further information on research design is available in the Nature Research Reporting Summary linked to this article.

## Data availability

Proteomic data generated in this study are available via ProteomeXchange with identifier PXD035119. They have been included also in Supplementary Data 1. The source data underlying the graphs and charts in the main manuscript are available as Supplementary Data 2. Uncropped gels have been included in the supplementary information. We have submitted all relevant data of our experiments to the EV-TRACK knowledgebase (EV-TRACK ID: EV220301)[72]. All other data from this study are available from the corresponding author upon reasonable request.

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

## Acknowledgements

We are grateful to Prof. Nunziatina De Tommasi for the fruitful discussions. This work is supported by a grant from the University of Salerno (FARB 2018-300390FRB18AM-BRO). M. Moros acknowledges funding from MCIN/AEI/ 10.13039/501100011033 and "ESF Investing in your future" (Grant RYC2019-026860-I).

## Author contributions

M.A., A.L. and A.A conceived the study; E.B., M.A., M.C. and A.A. set up the protocol for HR induction, EV purification and performed EV characterization. R.B and A.P. designed and carried out EV cell uptake and bioactivity studies in human cell lines. V.S and F.D.P performed proteomic and bioinformatic analyses. P.D.L and A.A. performed DLS analyses. M.M. carried out NTA and TEM experiments. All the authors participated in data analyses. AA wrote the manuscript. All the authors edited and approved the manuscript.

## Competing interests

The authors declare no competing interests.
