## [Peer Review File · Communications Biology]

Reviewers' comments:

Reviewer #1 (Remarks to the Author):

Boccia et al., in the manuscript entitled "Plant hairy roots as biotechnological platforms for the production of extracellular vesicles with therapeutic bioactivity", aimed at isolating extracellular vesicles from the hairy root cultures of *Salvia Dominica* and at evaluating the antineoplastic effects.

Despite the topic being interesting in my opinion the manuscript presents several limitations and should receive major revision before consideration for publication.

Below are my comments:

- The title of the manuscript is too generic for an original article, I suggest changing it to highlight the main result of the study.
- Also the abstract is too general; I suggest rewriting it to highlight the aim and the results of the study.
- The study aim at investigating a new platform for EV isolation. To validate this new approach, authors should isolate EV from the hairy roots of other plant matrices.
- The first part of the Results section (lines 156-161) should be transferred to the Materials and Methods section to avoid the duplication of the isolation protocol.
- After isolation, EVs were characterized through different techniques, however, the images of SEM analysis shown in Fig 2 c-d are not clear, I suggest the authors repeat the analysis and include more clear images.
- The major limitation of the study is the use of a single non-tumor cell line as well as of single pancreatic cancer cell line. The experiments presented in Figure 3 should be validated at least in another tumor cell line.
- The EV uptake could be performed at also early point, like 3-6 h, since it is known that EV internalization can occur after a few hours of incubation.
- Figure 3g-h-i is not mentioned in the result section, please add the figure reference in the text.
- The representative image of PARP-1 western blot is not clear in showing what the authors stated in the result section; please replace it with a new one showing the degradation of PARP
- The results about apoptosis showed in Fig 3g-i should also be evaluated in the normal cell line.
- The discussion is too long, some information can be moved to the introduction.

Minor comment:

- Line 260, correct the title of the paragraph "Bioactivity of HR-derived Extracellular Vesicles".
- A clinical trial using plant derived nanovesicles has been conducted showing safety and lowering of serum LDL in volunteers. It can be added between the reference (Raimondo et al, 2021).

Reviewer #2 (Remarks to the Author):

This is an interesting paper described a novel isolation system of plant derived vesicles from *Salvia dominica*. These vesicles have a good uptake to both HaCaT and MIA PaCa-2 cells, but interesting only influenced MIA PaCa-2 cells viability.

Authors claimed these vesicles from 10days culture medium are extracellular vesicles, but not dead cells or cell debris or even infected *Agrobacterium rhizogenes* from this wounding culture system. For this, they didn't provide any solid evidences. Actually, no EV markers shows abundantly in the proteomic data, but lots of proteins should be from nucleus. And to point out, this root hair culture system only applicable to very limited plant species. They have found an interesting phenomenon these vesicles only work to a cancer cell line, but not to another normal cell line. But they didn't provide any further investigations why this happened.

Here are specific comments:

1. The apoplast EV isolation method is relatively mature system now. You should compare the vesicle proteomic data from root hair system to apoplastic EVs and vesicles from cell debris to get a general idea how pure your vesicle preparations are.
2. The first step of the different centrifugations was 15.000g, dead cells in the medium will be

broken into small particles by this strength, there should be two steps before 15.000g, as 300g and 2.000g.

3. The different of the sizes between EM images/NTA and DLS suggested something wrong happened during experimental procedures. Fixation and negative staining shouldn't influence vesicle sizes that much.
4. At line 194, not sure the meaning to show this one-time concentration test.
5. Fig2.f shows a membrane debris, does author have any comments on that?
6. It's interesting that boiled vesicles don't have effects to MIA PaCa-2 cells. If the mechanism of the *Salvia dominica* is due to compounds as suggested in the discussion, dose these compounds destroyed after boiling? Does these compounds can't be uptake by the cells if not in vesicles? Does the concentration of the compounds are not effective if they are not in vesicles? Should have some discussion and investigation in this.
7. Fig3.a and b, looks like b have much more florescent EVs than a, even in the area between cells.
8. The discussion part is very broad and vague. For example, with only limited data for the treatment to one cancer cell line and one non cancer cell line can't have the conclusion as a safe profile for non-cancer cells (line 422).
9. Lots of typo in the manuscript need to be fix.

Reviewer #3 (Remarks to the Author):

In this manuscript, the authors isolated plant extracellular vesicles (EVs) from *Salvia dominica* hairy root. The authors showed the presence of EV-associated proteins using proteomic analysis. Moreover, the author also demonstrated selective pro-apoptotic activity in pancreatic cancer cells. However, this manuscript needs to add additional descript. I have a few comments the authors should address.

1. In the Figure 3, the authors assessed cellular uptake and viability test between non-cancer HaCaT cells and MIA PaCa-2 pancreatic carcinoma cells. Why did authors choose HaCaT cells to compare MIA PaCa-2 pancreatic carcinoma cells. It would be better to comment more specifically about explanation.
2. The authors evaluated MTT assay, and used EV preparation with heat-inactivated EV as a control. Authors should be better describe about heat-inactivated EV preparations in Materials and methods.
3. In the Figure 3a-d, the authors assessed cellular uptake test. The data showing fluorescence of annexin A1 appeared different intensity between Ctrl and Fluo-EVs.
4. In the Figure 3b, the authors treated Fluo-EVs in HaCaT cells, but the image did not appeared accumulated EVs into cytoplasm. Why did different intensity between HaCaT cells and MIA PaCa-2 cells. The authors need to add additional description.
5. The legend to Figure S5 of image should be corrected unit of scale bar.
6. In page 9, the subtitle requires correction (Bioactivity of HR-derived Cextracellular Vesicles).

POINT-TO-POINT REPLY

We would like to thank the reviewers for their valuable comments to improve our manuscript and for the opportunity to revise it. A summary of additional figures and changes to original figures precede our detailed reply to the reviewers' individual comments.

Our replies are **highlighted in red** with direct changes to the main text **highlighted in yellow**. Relevant figures and supplementary information have also been reproduced.

Summary of additional figures and changes to existing figures

Figure 2b. The NTA graph has been replaced with a new one including the last measurements

Figure 2c and d. The SEM images in c and d showing a group of EVs have been replaced with TEM images

Figure 2e. A new close up image of a single EV has been included

Figure 2g. The silver staining pattern has been replaced according to the new EV preparations

Table 1. The list of proteins has been replaced as new proteome analyses have been conducted on EV purified with the modified protocol

Figure 3 (a,b,c,d). Confocal images have been replaced after fluorescence normalization

Figure 3 i. The western blot analyses has been replaced with the images taken by new experiments as suggested by referees

Figure S3. A new NTA distribution plot replaced the former one.

Figure S4 (new). PCR analyses of *rolB*, *rolC* and *virD2* genes has been included to show the absence of *A. rhizogenes* contamination in EV preparations.

Table S1. The list of EV-associated proteins has been replaced with a new one according to the analyses conducted on the EV preparation after changing the purification protocol. Please note that this list contains proteins exclusively found in EV preparation and not in non-vesicular fractions, namely the pellets collected at 300 x *g*, 2000 x *g*, and 15000 x *g*

Figure S5. New Gene ontology analysis was performed on the EV-associated proteins. The new graphs have been included in this figure

Figure S9 (new). Apoptotic cell counts of HaCat cells were done according to the referee 1 suggestions

Figure S10 (new). MTT, FACS and western blot analyses were carried in MCF-7 breast cancer cell line in order to demonstrate the antitumor activities of HR-derived EVs in a second tumour cell line

Figure S11 (new). The impact of heat-inactivated EV was investigated also in on MCF-7 as negative control.

Supplementary dataset: this list contains all the proteins identified in the EVs, including those found in the non-vesicular fractions. We included these data to provide complete proteomic information generated by this work, but we did not use these data for discussion

Reviewers' comments:

Reviewer #1 (Remarks to the Author):

Boccia et al., in the manuscript entitled "Plant hairy roots as biotechnological platforms for the production of extracellular vesicles with therapeutic bioactivity", aimed at isolating extracellular vesicles from the hairy root cultures of *Salvia Dominica* and at evaluating the antineoplastic effects.

Despite the topic being interesting in my opinion the manuscript presents several limitations and should receive major revision before consideration for publication.

Below are my comments:

➤ The title of the manuscript is too generic for an original article, I suggest changing it to highlight the main result of the study.

AA: we changed the title to make it more specific:

Plant hairy roots of *Salvia dominica* as biotechnological platform for the production of extracellular vesicles with antitumor bioactivity

> Also the abstract is too general; I suggest rewriting it to highlight the aim and the results of the study.

AA: In the limits of the Abstract word counts indicated by the journal, we modified the abstract and included the aim and more results' details.

> The study aim at investigating a new platform for EV isolation. To validate this new approach, authors should isolate EV from the hairy roots of other plant matrices.

AA: we have already partially characterised EV from *Salvia sclarea* by Silver staining and TEM as well as by mass spectroscopy . We are preparing a new paper with the biophysical and functional characterization of these EVs. Therefore we send the following image as a confidential results for the reviewers.

A) *S. sclarea* HR culture. B and C) TEM images of EV purified from *S. sclarea* HR. D) Protein pattern of three EV preparations obtained from *S. sclarea* HR

> The first part of the Results section (lines 156-161) should be transferred to the Materials and Methods section to avoid the duplication of the isolation protocol.

AA: we removed the sentence (158-161) from the text and rewrote the EV purification method in materials and methods section. Similarly we removed lines 126-130 which appears redundant with the materials and methods section

> After isolation, EVs were characterized through different techniques, however, the images of SEM analysis shown in Fig 2 c-d are not clear, I suggest the authors repeat the analysis and include more clear images.

AA: We agree that the SEM images were low resolution. Unfortunately we could not have access to the SEM facility for technical problems. Therefore, we removed SEM images and performed additional TEM analyses on the new EV preparations purified with the revised purification protocol. According to the current guidelines we provided wide field and close up images of EVs by TEM. Here, we attached the current figure 3 with new TEM images (c and e) as well as new NTA and silver staining analysis

➤ The major limitation of the study is the use of a single non-tumor cell line as well as of single pancreatic cancer cell line. The experiments presented in Figure 3 should be validated at least in another tumor cell line.

AA: we agree with the reviewer and performed new experiments in MCF-7 mammary cancer cell line in order to strengthen our data. We included these new results in the supplementary figures 10 and 11. These new experiments showed that HR-derived reduce MCF-7 cell viability and trigger apoptosis according to FACS and western blotting analyses. Thanks to the reviewer's comment we are able to conclude now that *S. dominica* EVs exert similar effects in two cancer cell lines.

These results were included in the text as it follows:

To confirm the antitumor activity of *S. dominica* HR EVs, we evaluated their effects also in MCF-7 breast cancer cell line. As reported in supplementary figures S10 A and S10 B, 24 h and 48 h EV treatments reduce up to 60% the MCF-viability and promote programmed cell death in a dose-dependent manner, respectively. Selective Procaspase-3 and PARP-1 cleavage in MCF7 cells proved the activation of apoptotic molecular cascade upon 24 h EV treatment (figure S10 C). As expected, heat-inactivated EV did not affect MCF-7 viability (Fig. S11).

➤ The EV uptake could be performed at also early point, like 3-6 h, since it is known that EV internalization can occur after a few hours of incubation.

This is correct and we expect that EV enter earlier in cancer cells, as demonstrated in different in vitro and in vivo models. However, all the analyses conducted in this work have considered 24 h and 48 h as principal time points to check the bioactivity of EVs. The uptake mechanisms of plant EVs in human cells deserve great attention and we would like to focus on this aspect in a future work.

➤ Figure 3g-h-i is not mentioned in the result section, please add the figure reference in the text.

Now we mentioned Fig.3 g-h in the text.

➤ The representative image of PARP-1 western blot is not clear in showing what the authors stated in the result section; please replace it with a new one showing the degradation of PARP

AA. We changed the previous WB images. Besides the experiments already mentioned in the previous paper version, we performed additional western blot experiments to assess the pro-apoptotic activities of EVs in two cancer cell lines. Therefore, we changed the western blot image of PARP-1. Now the cleaved bands appear more clear in figure 3 i. We also included WB data on MCF-7 in figure S10. Here attached are the new figures.

Fig. 3: Uptake and biological effects of HR-derived EVs in human cells.

Figure S10. Effects of HR-derived EVs after 24 h and 48 h incubation in MCF-7 breast cancer cells

➤ The results about apoptosis showed in Fig 3g-i should also be evaluated in the normal cell line.

We are sorry for missing this data. We did FACS analyses on HaCat cells treated with EV. As reported in the new figure S10, EVs did not trigger significant apoptosis in control cell lines.

➤ The discussion is too long, some information can be moved to the introduction.

We shortened the discussion in order to make it more focused. Therefore we deleted some parts or moved them into the introduction and rephrased sentences.

Minor comment:

- Line 260, correct the title of the paragraph “Bioactivity of HR-derived Extracellular Vesicles”.

We fixed the typo

- A clinical trial using plant derived nanovesicles has been conducted showing safety and lowering of serum LDL in volunteers. It can be added between the reference (Raimondo et al, 2021).

We have now included the following sentence citing this article that supports the use of plant-derived nanovesicles in clinical practise:

More recently, the preliminary results of a pilot open label study demonstrated that EVs derived from *C. limon* juice exert positive effects reducing waist circumference and LDL cholesterol in healthy subjects(Raimondo et al., 2021).

Reviewer #2 (Remarks to the Author):

This is an interesting paper described a novel isolation system of plant derived vesicles from *Salvia dominica*. These vesicles have a good uptake to both HaCaT and MIA PaCa-2 cells, but interesting only influenced MIA PaCa-2 cells viability.

Authors claimed these vesicles from 10days culture medium are extracellular vesicles, but not dead cells or cell debris or even infected *Agrobacterium rhizogenes* from this wounding culture system. For this, they didn't provide any solid evidences. Actually, no EV markers shows abundantly in the proteomic data, but lots of proteins should be from nucleus. And to point out, this root hair culture system only applicable to very limited plant species. They have found an interesting phenomenon these vesicles only work to a cancer cell line, but not to another normal cell line. But they didn't provide any further investigations why this happened.

Here are specific comments:

1. The apoplast EV isolation method is relatively mature system now. You should compare the vesicle proteomic data from root hair system to apoplastic EVs and vesicles from cell debris to get a general idea how pure your vesicle preparations are.

This point is very important and according to referee #2 suggestions we changed the purification protocol (please check the point 2). Then, we compared the proteome of the EVs to that of the pellets collected at low speed centrifugation steps, which may represent cells and subcellular fractions different from EVs. We included the proteins selectively found in the EV fractions as short list in the main text (table 1) and complete list as supplementary table 1, while proteins commonly found in EV and non-vesicular fractions were included as Supplementary Dataset. Not surprisingly, EVs and low speed pellets share common proteins, as cells and subcellular fraction represent the source of EVs.

Interestingly we identified a specific subset of 146 proteins exclusively found EV preparations. As extensively discussed in the paper, and in the following points, HR-derived EV carry numerous proteins known to be associated with extracellular vesicles, including the tetraspanin TET-7. Nuclear proteins were not present anymore in the GO analysis.

The comparison between apoplastic vesicles and those released from hairy roots is a very important aspect and we believe it deserves a future investigation for the amount of work and time it may require. Current protocols of Apoplastic vesicles purification starts from leaves, therefore plant cultivation is necessary to get them. In this moment, this task will take a lot of time for us for following reasons:

- *S. dominica* is not a common officinal plant; it is very difficult to find *S. dominica* seeds on the market and usually we get a few of them from Saudi Arabia collaborators who collect seeds from *S. dominica* plants only in the proper season (the end of spring/beginning of summer). The cultivation of this species may require a few months.

- The protocols for apoplastic vesicles isolation from leaves are set up only in a few plant species. This means that we must first develop an appropriate protocol for *S. dominica* species and we cannot exclude it might require many attempts due to plant specific features (e.g a huge amount of hair on the leaf of *S. dominica* may affect apoplastic EV isolation)
- The development of a new protocol requires a sufficient amount of plant material. This means that we need months to grow up a significant number of plants and collect fresh leaves or plantlets necessary to make trials. This timing is not compatible with the priority to publish the novelty of our work based on the biotechnological platform that we propose for the first time.
- Last, but not least, we strongly believe that the comparison of two different classes of EVs (the apoplastic ones and those purified from Hairy roots) require many efforts and deserve a separate publication.

2. The first step of the different centrifugations was 15.000g, dead cells in the medium will be broken into small particles by this strength, there should be two steps before 15.000g, as 300g and 2.000g.

We thank the reviewer for this helpful comment. Accordingly, we changed the purification protocol by introducing these two additional steps (300g and 2000g speeds). We also collected the cell debris of different centrifugation steps, check their proteome and systematically compared them to that of the EVs. The new EV preparations, besides EV-associated proteins already mentioned in the first version of this manuscript, also carry the tetraspanin 7 (TET7). This protein shows high sequence similarity to TET8, which is currently considered a specific marker of plant exosomes.

We would like also to draw the reviewer's attention that we exclude *A. rhizogenes* contamination as we did not find *Agrobacterium* proteins in the EV protein dataset. However, we proved the absence of *A. rhizogenes* in EV preparation by PCR of *rolB*, *rolC* and *VirD2* genes. These data are now included in the new figure S4.

Figure S4. Absence of *rolB*, *rolC* and *virD2* genes in EV preparation checked by PCR. As positive control *A. rhizogenes* DNA has been used

3. The different of the sizes between EM images/NTA and DLS suggested something wrong happened during experimental procedures. Fixation and negative staining shouldn't influence vesicle sizes that much.

We did a new round of EV characterization after the changes of the purification protocol. For technical reasons we could not have access to the SEM facility. Therefore we repeated NTA and TEM analyses on the EV preparations obtained by . The new results confirm the previous size distribution with small fluctuation of the size range. However, NTA confirmed that the majority of EV (almost 82%) range between 100 and 200 nm, while almost 10% has a mean size lower than 100 nm

4. At line 194, not sure the meaning to show this one-time concentration test.

We are sorry for the mistake. The EV concentration was measured in three EV preparations by NTA. Each analysis was done in 5 technical replicates. The concentration of new EV preparations remained in the same range.

5. Fig2.f shows a membrane debris, does author have any comments on that?

Membrane debris could come from initial rupture of cells, but also for EV rupture during the purification, as often described for methods based on differential centrifugations (Guan et al 2020, [10.1021/acs.jproteome.9b00693](https://doi.org/10.1021/acs.jproteome.9b00693)). However, thanks to the changes in the purification protocol, now we are pretty sure that the cellular contaminants are removed through the initial low speed steps. To prove the absence of unexpected residues and/or EV rupture we included a new wide field image by TEM (Figure 2, attached) showing numerous EV with intact shape. We removed the figure 2f and added a new close up of an individual EV.

6. It's interesting that boiled vesicles don't have effects to MIA PaCa-2 cells. If the mechanism of the *Salvia dominica* is due to compounds as suggested in the discussion, dose these compounds destroyed after boiling? Does these compounds can't be uptake by the cells if not in vesicles? Does the concentration of the compounds are not effective if they are not in vesicles? Should have some discussion and investigation in this.

It is widely accepted that the perfect negative control for EV bioactivity does not exist. Sampling solutions, depleted media and destroyed EVs have been used to this purpose, but all of them have pros and cons.

The heat-inactivation is known to destroy the lipid bilayer of the EVs thus impairing their delivery function. Of course, boiling also affect the stability of many compounds shuttled by EVs. As the main purpose of this work is to provide the first report of plant EV purification from hairy roots, we will investigate this additional interesting points in future works.

If the reviewer believes that the use of this control might be confusing or if it does not add useful information to the EV bioactivity we can remove these tests.

7. Fig3.a and b, looks like b have much more florescent EVs than a, even in the area between cells.

AA. We normalised the fluorescence of all the images in the panel EV and changed the text accordingly

8. The discussion part is very broad and vague. For example, with only limited data for the treatment to one cancer cell line and one non cancer cell line can't have the conclusion as a safe profile for non-cancer cells (line 422).

We shortened the discussion and rephrased sentences in this section in order to make it more focused.

During the revision of the work, we tested the EV bioactivity also in MCF-7 mammary cancer cells confirming the antitumoral activity measured by means of MTT, FACS and Western blot of CASP-3 and PARP-1. We believe that the confirmation of antitumoral activity in a second cell line make more reliable the results and the conclusions on tumoral cells.

9. Lots of typo in the manuscript need to be fix.

We thank the referee, we have checked and fixed typos throughout the manuscript

Reviewer #3 (Remarks to the Author):

In this manuscript, the authors isolated plant extracellular vesicles (EVs) from *Salvia dominica* hairy root. The authors

showed the presence of EV-associated proteins using proteomic analysis. Moreover, the author also demonstrated selective pro-apoptotic activity in pancreatic cancer cells. However, this manuscript needs to add additional description. I have a few comments the authors should address.

1. In the Figure 3, the authors assessed cellular uptake and viability test between non-cancer HaCaT cells and MIA PaCa-2 pancreatic carcinoma cells. Why did authors choose HaCaT cells to compare MIA PaCa-2 pancreatic carcinoma cells. It would be better to comment more specifically about explanation.

We chose the HaCaT cells as they are routinely used by our group to test and compare the pharmacological activity of many compounds. Moreover, we also used this as a non-malignant cell line to test the cytotoxic activity of citrus-derived nano and microvesicles.

We changed the sentence in the text to justify the choice:

Firstly, we examined the EV cell uptake in the HaCaT cells (human keratinocytes), already employed by our group as non-malignant cells to test anticancer activity of plant-derived vesicles (Stanly et al., 2020), and MIA PaCa-2 pancreatic carcinoma cells, characterized by high invasiveness and drug-resistance (Belvedere et al., 2014).

2. The authors evaluated MTT assay, and used EV preparation with heat-inactivated EV as a control. Authors should better describe about heat-inactivated EV preparations in Materials and Methods.

We are sorry for that. The info are now included in Materials and Methods and attached below:

Cells treated with heat-inactivated EVs were used as negative controls. To this aim, EV preparations were boiled for 1 h, cooled down at room temperature for 2 h and then administered to the cell cultures.

3. In the Figure 3a-d, the authors assessed cellular uptake test. The data showing fluorescence of annexin A1 appeared different intensity between Ctrl and Fluo-EVs.

We checked this point and adjusted the fluorescence in Confocal images. Figure 3 (A-D) have been replaced.

4. In the Figure 3b, the authors treated Fluo-EVs in HaCaT cells, but the image did not appear accumulated EVs into cytoplasm. Why did different intensity between HaCaT cells and MIA PaCa-2 cells. The authors need to add additional description.

The reviewer is right. After fluorescence normalization, we change the sentence highlighting that MIA PaCa cells take up more EVs than non-malignant cells. We believe the dynamics of internalization and comparison between normal and cancer cells deserve future attention.

5. The legend to Figure S5 of image should be corrected unit of scale bar.

We are sorry for the mistake and adjusted the scale bar.

6. In page 9, the subtitle requires correction (Bioactivity of HR-derived Extracellular Vesicles).

We fixed the typo.

Reviewers' comments:

Reviewer #1 (Remarks to the Author):

Authors have addressed the criticisms raised by the reviewer. The paper is now in a publishable form

Reviewer #2 (Remarks to the Author):

1. Author are trying to prove vesicles from root hair culture system are EVs but not PDVs with proteome data. It's hard to have a proper control since the EV from root apoplast is not easy to get. So you tried to compare with the pellet from lower speed of the centrifugation. But actually if there is no contamination from the broken cells shed off the root, the lower speed of centrifugation are also pelleting medium or large EVs. Actually, authors should prove there are no dead cells shed of during the long culture days with this system.

2. Authors are trying to separate EVs from PDVs, actually there are some reviewer paper and research papers, some have proteome comparison, should cite these paper for discussion.

3. For the negative control as boiled EVs, how do you perform this? Didn't find in method part and the result part.

4. The figure for this paper need to improve as some of the figure is not clear, some are not good aligned together. And a lot of typo in manuscript.

Reviewer #3 (Remarks to the Author):

The author answered our question well, so we have no further comments.

POINT-TO-POINT REPLY

We are grateful to the reviewers for the time and expertise dedicated to the peer-review process of this manuscript. Our replies are highlighted in red with direct changes to the main text highlighted in yellow.

Reviewer #1 (Remarks to the Author):

Authors have addressed the criticisms raised by the reviewer. The paper is now in a publishable form

We thank once again the referee for her/his comments and suggestions, which have contributed to improve the manuscript.

Reviewer #2 (Remarks to the Author):

1. Author are trying to prove vesicles from root hair culture system are EVs but not PDVs with proteome data. It's hard to have a proper control since the EV from root apoplast is not easy to get. So you tried to compare with the pellet from lower speed of the centrifugation. But actually if there is no contamination from the broken cells shed off the root, the lower speed of centrifugation are also pelleting medium or large EVs. Actually, authors should prove there are no dead cells shed off during the long culture days with this system.

We understand that the setup of a quite new biological system, such as purification of EVs released from roots, needs an accurate evaluation of multiple aspects to ensure purity and reliable protocols. The presence of undesired cellular contaminants may be an issue, of course, as highlighted by the reviewer, in our experiments, and, in more general terms, in all the experimental work aimed at ascertaining the function of released vesicles from animal and plant systems.

We would like to clarify this point here:

-The presence of dead cells is a physiological phenomenon in cell and tissue cultures, both in plants and in animal systems. It may represent really a problem if a significant percentage of death cells is present. In order to provide quantitative data, we have counted, with a Burker chamber the number of floating (alive or dead) cells and determined their density. We found approximately only 30 floating cells/mL of *S. dominica* conditioned medium after three days of subculturing and a comparable number of floating cells was observed after 7 days.

-Considering this very poor floating cell density, and the biomass of HR (at least 2 grams containing hundred thousands cells), the overwhelming majority of vesicles are, therefore, originated by secretion from the hairy roots in the medium, through mechanisms that need to be elucidated in the future. Only a small fraction could be vesicles accidentally created by tissue ruptures or floating dead cells. We included these considerations in the discussion as reported below.

-We would like also to underline that, based on the previous recommendations of reviewer 2, we have already changed the purification protocol introducing an additional step to eliminate dead cells and we have repeated all the biological analysis on the effects on control and tumour cell lines, which have confirmed our previous observations.

We would also like to add as last technical note that during the handling of supernatants we did not disturb the pellets of low-speed centrifugation steps to avoid EV contamination.

Finally, we have added a comment in the paper on this:

“Taken together these data indicate that HR-released vesicles are true plant EV, however the presence of a small number of vesicles generated by accidental rupture of soft structures (e.g. root hairs) in the growth phase cannot be completely excluded”.

2. The same reviewer suggested us to compare the proteome of the EV preparation to those of the apoplastic vesicles and to the pellets collected at lower speed centrifugation steps. This last comparison was very informative to identify proteins specifically associated to the small extracellular vesicles (100-200 nm), such as TET7 protein with a high similarity to TET8, which is actually considered, together with PEN1, a specific marker of true extracellular vesicles in plants.

3. Authors are trying to separate EVs from PDVs, actually there are some reviewer paper and research papers, some have proteome comparison, should cite these paper for discussion.

We thank the reviewer for this helpful suggestion. We included a new important reference and highlighted this point in the discussion. Here the new sentence we added to the manuscript:

*We also noticed in the HR-released EV proteome the absence of the category “endoplasmic reticulum” (Fig. S5 A). According to a recent comparative proteomic study of plant-derived nanovesicles and small EVs in *A. thaliana*, this could be a distinctive feature of the true EV proteome⁵⁵.*

4. For the negative control as boiled EVs, how do you perform this? Didn't find in method part and the result part.

Heat-denatured EV have been extensively used in the literature as negative controls. The preparation of the negative control is included in the MTT paragraph :

To this aim, EV preparations were kept at 95 °C for 1 h, cooled down at room temperature for 2 h and then administered to the cell cultures.

In the results, we included the following sentences and appropriate citations:

Additionally, both cell lines were incubated with heat-denatured EV preparations, which have been previously used as negative controls⁴⁸⁻⁵⁰

5. The figure for this paper need to improve as some of the figure is not clear, some are not good aligned together. And a lot of typo in manuscript.

We aligned better the images in the figure 1. The images in figure 3 are not aligned as they are odd in number. If this is a problem for the journal, we will discuss the point with the Check Quality Officers.

We also removed the typos and added small changes to the text (all highlighted in yellow). Typos revision has been made also in the Supplementary Information.

Reviewer #3 (Remarks to the Author):

The author answered our question well, so we have no further comments.

We have appreciated very much your contribution to improve the paper and make it more readable

REVIEWERS' COMMENTS:

Reviewer #2 (Remarks to the Author):

The author answered our question well, so we have no further comments.